# Machine Learning-Based Assessment of Parkinson’s Disease Symptoms Using Wearable and Smartphone Sensors

**DOI:** 10.3390/s25164924

**Published:** 2025-08-09

**Authors:** Tomasz Gutowski, Olga Stodulska, Aleksandra Ćwiklińska, Katarzyna Gutowska, Kamila Kopeć, Marta Betka, Ryszard Antkiewicz, Dariusz Koziorowski, Stanisław Szlufik

**Affiliations:** 1Faculty of Cybernetics, Military University of Technology, gen. Sylwestra Kaliskiego 2, 00-908 Warsaw, Poland; 2Department of Neurology, Faculty of Health Sciences, Medical University of Warsaw, Żwirki i Wigury 61, 02-091 Warsaw, Polandstanislaw.szlufik@wum.edu.pl (S.S.)

**Keywords:** Parkinson’s disease, machine learning, symptom severity, wearable sensors

## Abstract

This study explores the use of machine learning models to assess the severity of Parkinson’s disease symptoms based on data from wearable and smartphone sensors. It presents models to predict the severities of individual symptoms—tremor, bradykinesia, stiffness, and dyskinesia—as well as the overall state of patients, using both clinician and patient self-assessments as labels. The dataset, although limited and imbalanced, enabled the identification of key trends. The best performance was achieved when combining data from both the MYO armband and smartphone, and when using patient self-assessments as targets. Tremor was the most predictable symptom, while others proved more challenging—especially at higher severity levels, which were poorly represented in the dataset. These results highlight the value of multimodal data and the importance of patient input in symptom monitoring. However, they also point to the need for more balanced and extensive datasets to improve prediction accuracy across all severity levels and symptoms.

## 1. Introduction

Parkinson’s disease (PD) is one of the most common neurological disorders [1], significantly impacting patients’ quality of life and making daily tasks increasingly difficult. The disease manifests through both motor and non-motor symptoms, each of which requires different approaches for treatment [2,3].

Motor symptoms are the most widely recognized features of PD. The characteristic resting tremor, typically observed when muscles are at rest, is a defining symptom that presents as repetitive limb trembling, with a frequency between 4 and 6 Hz [2]. Bradykinesia, another primary motor symptom, represents the slowness of voluntary movement [2]. It is marked by longer action times and requires patients to exert greater effort and focus to perform movements. Additionally, muscle rigidity or stiffness is common, further complicating movements, causing pain, and reducing the range of motion [2]. These are the three core motor symptoms—tremor, bradykinesia, and rigidity—experienced by the patient due to the presence and advancement of the disease. More complex motor issues, such as freezing of gait, balance instability leading to falls, handwriting difficulties, and voice disorders, may manifest in advanced stages or in a smaller subset of patients [3].

Apart from motor symptoms, patients also suffer from a diverse range of non-motor symptoms encompassing physiological and psychological issues [2]. Examples include depression, lack of emotional involvement, sleep problems, and constipation. With the advancement of the disease, these symptoms might become more troublesome than the motor symptoms [2].

The cause of the disease is the progressive degeneration of dopaminergic neurons responsible for producing dopamine, which is a neurotransmitter related to regulating movement and various neurocognitive functions [4]. Dopamine deficiency is the main cause of PD symptoms, and the treatment strategies focus on restoring dopamine levels or enhancing the brain’s sensitivity to this neurotransmitter. Levodopa, a dopamine precursor capable of crossing the blood–brain barrier, remains the most effective medication for managing PD symptoms [5]. Usually, it is administered orally using pills, though alternative delivery methods such as duodenal levodopa infusion (Duodopa) are available for advanced cases [6].

However, many long-term levodopa users can face additional complications, such as dyskinesias—rapid, involuntary movements caused by fluctuations in dopamine levels [5]. As the disease progresses, the therapeutic window narrows, complicating medication management and necessitating highly personalized treatment regimens [3,5].

The complexity of PD management requires individual, patient-specific plans aimed at symptom mitigation to improve quality of life [7]. In order to find the optimal medicine doses, the clinicians assign an initial medication schedule that is then adjusted based on the patient’s individual reports [5,7]. However, these adjustments are heavily dependent on subjective patient reports, which are often imprecise and subject to variability. This introduces significant challenges in optimizing treatment, underscoring the urgent need for objective and reliable methods to monitor and quantify PD symptoms. Such a comprehensive assessment should encompass tremor, bradykinesia, rigidity, dyskinesias, and an overall evaluation of the patient’s condition.

Over the years, various approaches have been developed to objectively assess PD symptoms [8,9]. These methods often employ algorithms to detect and quantify specific symptoms or leverage machine learning models to evaluate the patient’s clinical state [10,11,12,13,14,15,16]. Traditional algorithmic approaches have typically focused on identifying the presence of symptoms or, in more advanced cases, classifying their severity into predefined categories. However, these systems have several limitations: they often rely on data from a single sensor or a specific exercise, making it difficult to generalize findings across different sensor types, tasks, or symptom manifestations. An example of such an approach was proposed by Griffiths et al. [14]. They created an algorithm which, based on continuous accelerometer readings, provides a score for both bradykinesia and dyskinesia severity throughout the day. Other studies focused on using machine learning models for predicting symptom severity. During the DREAM Challenge [12], and in a study by Gutowski [17], both shallow machine learning and advanced deep learning models were developed to predict the severity and presence of three individual symptoms, tremor, bradykinesia, and dyskinesia, also based on accelerometer readings from different limbs. A study published by Thomas et al. [13] focused on predicting a universal value—a treatment–response index, which was designed to represent the symptoms that could be captured using accelerometers and gyroscopes and reflect the response to treatment.

The presented solutions using machine learning were only able to detect the presence of the symptom or, in more sophisticated cases, classify its intensity into one of the predefined severity classes. Furthermore, these approaches usually focused on using data from a single sensor or a single exercise, making it difficult to compare results across different sensors, tasks, and symptoms. The study presented in this paper focuses on experiments aimed at building prediction models for evaluating the severity of four main symptoms associated with PD, tremor, bradykinesia, muscle stiffness, and dyskinesia, as well as the overall patient state. In contrast to previous work, it aims to build models capable of predicting a real-valued score on a 0–4 scale representing symptom severity. This enables fine-grained prediction for all considered symptoms and the overall condition based on multimodal sensor data collected during diverse motor tasks. Unlike prior studies focusing on binary classification or severity categories, this work provides continuous severity estimates and evaluates both clinician- and patient-based labels, facilitating future research into subjective versus objective symptom assessment. This approach offers more nuanced feedback for monitoring disease progression and treatment efficacy and can support personalized adjustments to treatment regimens.

## 2. Materials and Methods

### 2.1. Dataset

The dataset used in this research was created as a result of cooperation between two research facilities, the Military University of Technology (gen. Sylwestra Kaliskiego 2, 00-908 Warsaw, Poland) and the Medical University of Warsaw (Żwirki i Wigury 61, 02-091 Warsaw, Poland). The dataset consists primarily of recordings from patients with PD. It is the outcome of a study on the use of a mobile application in the differential diagnosis and treatment of tremor in patients with essential tremor, PD, and atypical parkinsonism. This study was approved by the Bioethics Committee of the Medical University of Warsaw. During this study, data collection was initially supervised by clinicians. The process was supported and organized by an information system, which consists of a mobile application and a web portal. The data was mostly collected using the mobile application.

The mobile application is designed to collect patient demographic and clinical data upon registration of the patient. However, its main goal is to allow evaluation of the patient’s state and track state changes, medicine schedules, and intakes. It allows for the conducting of examinations, during which the patient engaged in a series of exercises aimed at capturing different symptoms of the disease. These were performed using the mobile device and wearable sensors, including the Myo armband [18]. The application supports the collection of data during four types of exercises: sensor, reaction, handwriting, and speech exercises. However, this paper focuses solely on the severity assessment of individual symptoms based on the signals recorded during sensor exercises.

The main goal of sensor examinations is to detect motor symptoms of PD, particularly tremor, bradykinesia, and dyskinesia. To do this, the application enabled data collection with built-in sensors in the mobile device—with the accelerometer and gyroscope at a frequency of 50 Hz, and sensors from the wearable device—an accelerometer, gyroscope, and EMG data from the MYO armband. Before starting the examination, the patients were asked to put on the wearable sensors (if applicable) on one or two arms and hold the mobile phone in the examined hand.

The first sensor task was focused on detecting rest tremor. The patient was asked to keep their hands on their knees or a vertical platform for 30 s while the sensor data was collected. The second task was focused on detecting postural tremor—for 30 s, the patient extended their arms in front of them to record the data with sensors. Next, the patient performed a 30 s pronation–supination task—primarily for detecting bradykinesia. The sensor data were collected for another 30 s while the patient performed further tasks. This task aimed to assess the kinetic tremor experienced by the patient.

After each examination, performed under clinical supervision, the clinician evaluated the four main symptoms associated with PD on a scale from 0 to 4: bradykinesia, tremor, dyskinesia, and muscle stiffness. Additionally, both the clinician and the patient provided separate evaluations of the overall patient state (representing the response to medication), using a scale from −4 (severe symptoms) to 4 (severe dyskinesia), with 0 representing the optimal state. This evaluation allows for the capturing of the overall response to medication and can be later used to evaluate and adjust treatment.

A neurologist, Stanisław Szlufik, from the Mazovian Bródno Hospital (Ludwika Kondratowicza 8, 03-242 Warsaw, Poland), along with his team, was responsible for providing state evaluations, which were treated as the ground truth for experiments described in the paper. At the time of data analysis, this dataset contained accounts of 241 patients with PD, resulting in 739 examinations. However, not all the examinations included all exercises for two main reasons: first, the scope of the scales and assessments evolved over time; and second, to better capture the scope and magnitude of Parkinson’s disease symptoms, clinicians were allowed to restrict the set of exercises for each examination. The characteristics of the dataset are presented in Table 1.

### 2.2. Data Preparation

The dataset contains a limited number of samples; therefore, deep learning methods—despite their popularity in modern research—did not yield significant results. Previous work by Gutowski [17,19] involved the development of deep learning models using a significantly larger dataset, which resulted in strong performance. However, when these models were applied to the current, much smaller dataset—including attempts with transfer learning—the results were significantly worse than those obtained using shallow models. This motivated the decision to exclude deep learning approaches from this manuscript and focus solely on conventional, shallow ML models, which require additional preprocessing and the extraction of relevant features from raw signal data. The process of preparing the raw sensor signal for ML training and prediction is presented in Figure 1.

The raw signal was processed through actions such as filtering to remove unwanted components from the raw signal, calculating the magnitude of the signal, and decomposing it into multiple signals. This was then followed by the feature extraction step. Based on the signal type and the purpose of the model (what variable is predicted), a set of features was selected. These features were calculated based on the signal and should represent the signal well, given the model’s prediction task. If the number of created features is high, appropriate methods are often employed to reduce the number of variables, leaving only those most relevant. The reduced number of features can then be delivered as the inputs to the ML model.

The signals from inertial sensors needed some preparation for the ML models. They were first filtered using a high-pass filter in order to remove the gravitational acceleration component, with a cut-off frequency of 0.1 Hz. All of the signals were also filtered with a low-pass filter to remove all noise. Since the sampling frequency was 50 Hz, a low-pass filter with a cut-off frequency of 20 Hz was selected. PD-related tremors and other motor features do not typically exceed this frequency. For filtering the signals, the Butterworth filter [20] was used, which is often utilized for its flat frequency response in the passband, ensuring minimal signal distortion before the cutoff frequency. This approach balances the removal of unwanted components and retaining crucial movement data, facilitating accurate feature extraction and analysis.

After filtering, the signals were used to calculate the magnitude signal (1), representing movement captured in all directions. This derived signal allows for a more aggregated analysis, disregarding the direction of movement, which might be important, especially in cases where the sensors are not always worn in the same orientation.(1)$$M=\sqrt{X^{2}+Y^{2}+Z^{2}}$$

### 2.3. Feature Extraction

Based on the description of the main PD symptoms—dyskinesia, bradykinesia, and tremor—and consultations with neurologists, the signal was decomposed into three frequency bands: 0–3, 3–9, and 9–14 Hz. The features were then calculated for each of these frequency bands and the whole signal, for each axis, and the magnitude signal. This allowed for the capturing of different aspects of the signal, providing an accurate and precise representation. The features selected to be calculated were chosen based on a literature review [11,13,21,22,23,24,25,26] regarding the analysis of inertial signals for detecting activities, diagnosing PD, and quantifying PD symptoms.

These features were divided into time domain features, calculated directly from the signal, and frequency domain features, which provided insights into the signal’s frequency content. To extract frequency domain features, the signal underwent a Fourier Transform [27], a process that decomposes the signal into its constituent frequencies, revealing the spectrum of frequencies present and their relative intensities. Part of this analysis involved computing the Power Spectral Density (PSD), which quantifies the power present within each frequency component of the signal. The PSD is crucial for understanding the energy distribution across various frequencies, enabling the identification of dominant frequency bands that may indicate the presence of tremors or other PD-related motor symptoms. This helped highlight the specific frequencies contributing to the signal and aided in detecting patterns or abnormalities in the frequency domain, offering a better representation of how PD affects motor functions. Table 2 contains a list of features that are calculated based on the signal for each axis and the magnitude.

The features were calculated using functions from the NumPy (v1.24.4), SciPy (v1.10.1), PyWavelets (v1.4.1), and EntropyHub (v2.0) Python (v3.8.10) libraries. Additional custom features were individually implemented in Python. Time domain features were calculated for the entire signal, across all three axes, and the signal magnitude, resulting in 44 features. Frequency domain features were computed for 3 previously described frequency bands and the original signal across all axes (X, Y, Z, and magnitude), yielding 128 features. To ensure that the signal’s characteristics were captured as accurately as possible, additional features were added.

A short-time Fourier transform (STFT) was performed with a window size of 4 s and a 2 s overlap. For each window, the mean PSD was calculated, and the following statistics were computed for the vector: the mean, standard deviation, skewness, min, and max. This resulted in 5 additional features for each axis and each frequency band, totaling 80 features. Similarly, the raw signal was segmented into windows of this size. For each window, the value range and the entropy were calculated, as described by E. Sejdić et al. [22]. Based on these values, the previously described statistics were calculated, adding 40 more features.

Following the methodology described by Thomas et al. [13], a three-level Discrete Wavelet Transform was applied using a Daubechies wavelet of order 10. The means and the standard deviations were calculated for first-level high-frequencies, second-level high-frequencies, and third-level high-frequencies. These calculations resulted in an additional 24 features.

To capture the correlations between different axes, Pearson correlation coefficients were calculated for each axis pair (*X* and *Y*, *X* and *Z*, and *Y* and *Z*), resulting in three features. In total, 275 features were extracted from a single accelerometer signal.

### 2.4. Examination Metadata

To build appropriate models for predicting the patient state, alongside the features extracted from the collected sensor data, additional features were added to improve the quality of the model and its prediction precision. These features were patient characteristics equal among all examinations of that patient, as well as characteristics of specific examinations. The full list is showcased in Table 3.

For categorical features, such as the affected side, handedness, and groups, one-hot encoding was applied to ensure correct interpretation of the values by ML models. The remaining features were normalized along with sensor-derived features by subtracting the mean and dividing it by the standard deviation.

### 2.5. Feature Selection

A single sensor examination performed by a patient can consist of 3 exercises. When the examination is performed for both hands, the number can increase to 6. Considering the number of sensors and extracted features, one examination can provide thousands of features, a number that can be easily higher than the number of patients and even the number of total examinations performed. As stated by Guyon and Elisseeff [29], in these situations, it is important to consider feature selection methods. These can reduce the number of dimensions and therefore make it easier for the ML model to learn the dependencies in data, as well as perform the training process faster. Recent advances in feature selection, such as the multi-objective binary grey wolf optimization with guided mutation [30], demonstrate how intelligent optimization techniques can further enhance feature selection efficiency and model performance.

The process to restrict the number of features was performed in two steps. The first step focused on removing the variables that are highly correlated with each other. Having duplicate features does not improve the performance of ML models but only slows down the process. Therefore, the Pearson’s correlation coefficient was calculated for every pair of features in question. Whenever there was a correlation value above 0.97 between two features, these were excluded from further analysis.

The second step in the reduction in feature dimensions was applied during the training process. First, all features were assigned importance scores using the Random Forest [31] ML model. They were then sorted in descending order, and only the top 60% were further considered. Feature selection was then performed through the following process: Beginning with the most important feature, additional features were added one by one. At each step, the model’s cross-validated performance was evaluated, and a feature was retained only if it led to an improvement.

This method was selected because, even after the initial step, some features were highly correlated, which can negatively impact methods that evaluate features individually, such as permutation importance [32], since permuting one feature does not fully remove shared information with correlated features. Moreover, methods such as Principal Component Analysis (PCA) [33] and scikit-learn’s SelectFromModel [34] did not improve model performance in the experiments. The iterative, performance-driven feature selection, combined with initial removal of highly correlated features, proved to be more effective, enabling the construction of quality models that provided accurate predictions based on as few as 30 features.

### 2.6. ML Model Training

Three different machine learning models from the scikit-learn (v1.2.2) and XGBoost (v2.0.3) Python libraries were selected for regression: Random Forest [31], Extreme Gradient Boosting (XGBoost) [31], and Support Vector Machine (SVM) [35]. These models were initialized with default parameters.

Random forest (RF) [31] is an ensemble method that bases its decisions on multiple individual ML models, such as decision trees. Each tree is trained on a random subset of the data and features, and the final prediction is made by combining the predictions of all trees. This approach reduces overfitting and improves generalization compared to a single decision tree.

XGBoost (XG) [31] is also an ensemble model that uses a collection of decision trees. However, it builds these trees sequentially, where each tree aims to correct the errors of the previous ones. This method, known as boosting, enhances the model’s accuracy and robustness by focusing on the mistakes made by prior models, thereby improving predictive performance.

SVM [35] can be used both for classification and regression tasks. For classification, it tries to find the optimal hyperplane that best separates members of different classes in the feature space. SVM supports the use of kernel functions, which can transform the data into a higher-dimensional space, making the separation process easier and enabling the handling of non-linear boundaries. The default kernel in scikit-learn is the Radial Basis Function [35].

To evaluate the performance of regression models, defined metrics are calculated on the prediction results of the test set. Commonly used metrics in evaluation are the mean squared error (*MSE*) (13), the coefficient of determination (*R*^2^) (14), the mean absolute error (*MAE*) (15), and Pearson’s correlation coefficient (*r*) (16) between the true values and predicted outcomes [36]. These give a good overview of the overall performance of the machine learning models. However, in cases of highly imbalanced datasets, such as this one, these metrics might not give enough information for model evaluation. Therefore, two additional metrics were constructed. Since the original labels are discrete values, which were considered classes previously, it was possible to calculate class-specific metrics. The *MAE* was selected to be calculated for every class separately (17). This was used to create a derived metric, *bMAE* (18), which represents the mean absolute error across different classes; similarly, *bMSE* (19) was defined using the mean squared error calculated for every class.(13)$$MSE=\frac{1}{n}\sum_{i=1}^{n}{\left(y_{\mathrm{true},i}-y_{\mathrm{pred},i}\right)}^{2}$$(14)$$R^{2}=1-\frac{\sum_{i=1}^{n}{\left(y_{\mathrm{true},i}-y_{\mathrm{pred},i}\right)}^{2}}{\sum_{i=1}^{n}{\left(y_{\mathrm{true},i}-\bar{y_{\mathrm{true}}}\right)}^{2}}$$(15)$$MAE=\frac{1}{n}\sum_{i=1}^{n}\left|y_{\mathrm{true},i}-y_{\mathrm{pred},i}\right|$$(16)$$r=\frac{\sum_{i=1}^{n}\left(y_{\mathrm{true},i}-\bar{y_{\mathrm{true}}}\right)\left(y_{\mathrm{pred},i}-\bar{y_{\mathrm{pred}}}\right)}{\sqrt{\sum_{i=1}^{n}{\left(y_{\mathrm{true},i}-\bar{y_{\mathrm{true}}}\right)}^{2}\sum_{i=1}^{n}{\left(y_{\mathrm{pred},i}-\bar{y_{\mathrm{pred}}}\right)}^{2}}}$$(17)$${\mathrm{MAE}}_{k}=\frac{1}{n_{k}}\sum_{i\in \mathrm{class}\;k}\left|y_{\mathrm{true},i}-y_{\mathrm{pred},i}\right|$$(18)$$bMAE=\frac{1}{C}\sum_{k=1}^{C}{\mathrm{MAE}}_{k}$$(19)$$bMSE=\frac{1}{C}\sum_{k=1}^{C}\;\frac{1}{n_{k}}\sum_{i\in \mathrm{class}\;k}{\left(y_{\mathrm{true},i}-y_{\mathrm{pred},i}\right)}^{2}$$

### 2.7. Individual Symptom Evaluation

The features derived from signals collected during patient exercises represent the patient’s condition during the examination. The features provide different aspects of the examination performance and might be important in identifying specific symptoms of PD. At the end of examinations conducted in the presence of the clinician, a state assessment screen is displayed, where the overall state evaluation is provided along with individual symptoms, including tremor, bradykinesia, muscle stiffness, and dyskinesia. The clinician is asked to evaluate their severity on a scale of 0 (not present) to 4 (very severe). While this evaluation was not provided in all examinations for PD patients, 356 of the patient examinations contain these evaluations. This section focuses on building ML models capable of predicting individual symptom severities (as evaluated by clinicians) based on exercise-derived features.

In this dataset, the problem of imbalance is significant. The total number of samples is low, and higher symptom severities are poorly represented. For example, there is only one sample for dyskinesia severity of 4, making it impossible to train and evaluate the model for this severity. Other symptoms have better representation, with the most balanced dataset being for tremor prediction—10 samples for a severity of 4. The class distributions for all symptoms (tremor, bradykinesia, muscle stiffness, and dyskinesia) are shown in Figure 2.

Due to the small number of examinations in the dataset, to validate the ML models, cross-validation [37] was employed. It is a technique where the data is randomly split into k disjoint sets. The training process is then performed and evaluated k times, with k − 1 subsets used for the training and the remaining subset used for evaluation. This process is repeated k times, ensuring that every subset is treated as the test set exactly once.

In the simplest version of cross-validation, splitting into subsets is performed randomly. However, there are more advanced versions that can be used for specific scenarios. For example, stratified k-fold cross-validation is often used for classification problems. In this method, the partitioning is performed so that the distribution of class samples in different subsets is similar.

Additionally, Leave-One-Out (LOO) cross-validation can be used. It can be performed either on individual samples or on groups. When performed on samples, each subset contains only one sample. When performed on groups, the number of subsets is equal to the number of groups, with each model evaluated on one group while being trained on the remaining groups.

These splits are included in the scikit-learn Python library in the form of the following classes: KFold, StratifiedKFold, LeaveOneOut, and LeaveOneGroupOut. These are used to perform the training process in this part of the study.

During the training process, numerous training processes are executed; they can be grouped into three groups based on the expected goal of the training:Single exercise, single sensor from one device—finding which sensor and device combination best captures specific symptoms during different exercises,Single exercise—finding which exercise is best at capturing each of the symptoms,All exercises and devices—finding out how the models perform at capturing symptoms when all of the data can be used.

Each experiment was performed using all of the previously defined models. Each experiment was validated using cross-validation with two different splits—10-fold split (10F) and leave one patient out (LOO)—to see how models perform in these different situations.

### 2.8. Overall State Evaluation

The comprehensive assessment of a patient’s overall state plays an important role in understanding the nature of PD. While detailed evaluations of specific symptoms offer valuable insights into the disease’s characteristics, severity, and symptom manifestations, they may not fully encompass the impact on a patient’s quality of life and daily functioning. To address this, the MDS-UPDRS [38] provides a foundational framework for a more inclusive evaluation. In an effort to simplify the case and represent the therapeutic effect of medication, Westin et al. [39] proposed the TRS scale, optimizing it to capture the spectrum of patient experiences from severe symptoms to severe dyskinesia, with 0 being the optimal state. The TRS scale used in this study ranges from −4 to +4, as presented in Figure 3. It was adjusted to allow clinicians to gauge the overall state of PD patients more effectively. Such a comprehensive assessment is crucial for monitoring disease progression and customizing treatment plans to align with the dynamic needs of each patient, thereby enhancing therapeutic outcomes and patient well-being.

In this section, the focus is on the development of machine learning models capable of predicting the adjusted TRS scale values. The predictions are based on a set of data collected during patient evaluations. These include sensor exercises, screen interactions, handwriting, and vocal exercises. By analyzing a diverse collection of examination data, the models aim to achieve a more accurate and personalized understanding of patient conditions. This approach is designed to enhance the precision of treatment plans, tailoring interventions to meet the unique needs of individuals with PD.

The goal of training ML models is to evaluate the patient’s state during examinations. The ground truth values for this were provided both by the patient—their subjective opinion—and by their clinician, which is hopefully, more objective.

While the models were trained separately on clinician- and patient-reported labels; no direct comparison between these two groups was performed in this study. However, the predictions based on these two sources showed noticeable differences, which is likely caused by the nature of the input. Clinician assessments tend to reflect standardized diagnostic criteria and are influenced by medical training and experience, whereas patient self-reports are subjective and may be shaped by individual perceptions or mood. Previous research [40,41] has also highlighted discrepancies between patient- and clinician-reported outcomes, especially in conditions involving fluctuating or non-visible symptoms of the disease.

As for the individual symptom severities, this dataset was also affected by an imbalance in the label values. Furthermore, the range of values is more than twice as big, and the precision is higher, which is shown in Figure 4 along with the number of examinations that had each of the labels assigned. This makes it more difficult to prepare a model that performs well in the range of values.

The process to build ML models for predicting the state of the patient is similar to the prediction of specific symptom severities. Regression models were built using sensor signals registered from different exercises; similarly, the 10-fold and leave-one-patient-out cross-validation was performed, and the previously described metrics (*R*^2^, *r*, *MAE*, *bMAE*, *bMSE*) were used to evaluate the models. The scope of experiments is the following:Single device—finding which device is more useful in capturing the scope of the disease,All collected data—building a complete and optimal model for predicting a patient’s state.

Due to the larger number of possible values than for the symptom severities, the results are presented in the form of a scatterplot instead of a violin plot. It is used to present machine learning regression results by plotting the true values on the *x*-axis and the predicted values on the *y*-axis, allowing for the assessment of the model’s performance.

## 3. Results

### 3.1. Individual Symptom Evaluation

The primary goal of the initial training process was to evaluate symptom severities based on data from individual exercises and sensor signals. For each training instance, all previously discussed machine learning models were applied and configured to address the regression task of estimating the severity of tremor, bradykinesia, muscle stiffness, and dyskinesia.

Model performance was assessed using standard evaluation metrics, and the top-performing models (those achieving the highest *R*^2^ scores) are summarized in Table 4, with complete results provided in Appendix A. The table lists two models per symptom: one trained with 10-fold cross-validation (10F) and the other using leave-one-patient-out cross-validation (LOO). Comparing these models helped assess how individual patient characteristics influence model performance.

Table 5 and Table 6 extend this analysis by evaluating symptom severity predictions using all sensor signals collected during a single exercise (Table 5) and the full dataset comprising all sensor data recorded during the examination (Table 6).

The results clearly show that tremor severity can be predicted with the highest accuracy, as confirmed by the Wilcoxon signed-rank test [42] (e.g., received a *p*-value of 1.1 × 10^−6^ when compared with bradykinesia). This is expected, as tremor is one of the most prominent and easily observable symptoms of PD. It is then followed by bradykinesia, likely due to its characteristic slowness of movement being relatively easy to detect through time-series sensor data. In contrast, dyskinesia prediction performed the worst, which is consistent with its limited representation in the dataset—particularly at higher severity levels, as shown in Figure 2.

The differences between 10F and LOO cross-validation results were generally small. To assess statistical significance, the Wilcoxon signed-rank test was again applied. With a test statistic of 1420.5 and a *p*-value of 0.436, no statistically significant difference was found at the 0.05 significance level. This suggests consistent model performance across different split approaches and indicates low susceptibility to data leakage—likely due to the diverse patient pool and the limited number of repeated examinations per patient.

From a task-specific perspective, exercises influenced performance differently depending on the symptom. Tremor and dyskinesia, which are observable through involuntary movement, were best captured during the first exercise, where the patient remained at rest. Conversely, bradykinesia and stiffness—requiring active motion—were best evaluated during the third exercise, which involved the pronation–supination task. Overall, accelerometer signals provided more informative features for severity prediction across all symptoms.

To visualize the prediction accuracy across severity levels, violin plots were created for models using all sensor data. These violin plots provide the distribution of predictions and class-specific *bMAE* values (Figure 5 and Figure 6).

The violin plots highlight several limitations of the models. First, they consistently struggle to predict higher severity levels: predictions rarely exceed a value of 3 when the ground truth is 4. Second, particularly for stiffness, a severity level of 0 is rarely predicted. This is a notable limitation, as correctly identifying the absence of symptoms is critical for clinical validity. One possible explanation is the imbalance in class distribution: for both bradykinesia and stiffness, level 1 was the class with the most samples. Furthermore, severity level 0 in these two symptoms often corresponds to extended periods of immobility, making it harder for the models to distinguish from low but non-zero symptom levels. Lastly, some models occasionally predict negative values, which do not occur in the original dataset. To solve this problem, one of the following approaches could be selected:Using post-processing to clip predicted values to the valid range [0, 4],Reformulating the problem as a classification task (ordinal classification). However, this would lead to a loss of prediction precision.If neural networks were explored, applying a bounded activation function scaled to the target range in the final layer of the model.

Considering the advantages and disadvantages of these methods, clipping the values to the valid range is the best solution for this problem.

Since the clinical deployment depends on the transparency of the models, further experiments were performed to assess the importance of specific features in the prediction of severity for each symptom. To calculate these importances, the permutation feature importance method [32] was used with *R*^2^ as the scoring metric. This method works by randomly shuffling the values of each feature and measuring the resulting decrease in the model’s performance. This enables identification of the features the model relies on most for accurate predictions—the greater the drop in performance, the more important the feature is considered. The top five most relevant features and their corresponding importance scores are presented in Table 7.

Based on the results presented in the table, distinct sets of features emerged as most relevant for detecting and estimating the severity of specific Parkinson’s disease symptoms. For tremor, the highest-ranked features were frequency-based parameters, particularly derived from gyroscope data on the *Z*-axis, such as the weighted mean power and spectral centroid within the 3–9 Hz and 0–25 Hz bands, respectively. These frequency bands align with the known physiological range of tremor in Parkinson’s disease. Notably, both MYO and smartphone sensors contributed top-ranking features, indicating that multiple modalities are effective for tremor characterization.

In the case of bradykinesia, the most important features were predominantly time-domain statistics, such as the skewness, mean, and median, computed from accelerometer signals. These features reflect the irregular and reduced amplitude of movement typically associated with bradykinesia. Interestingly, features from both hands and all three axes contributed, suggesting that bradykinesia manifests in a more globally distributed motor pattern.

Dyskinesia and stiffness also showed distinct profiles. For dyskinesia, which involves involuntary, excessive movements, frequency-domain features again dominated, especially spectral power and related descriptors like the interquartile range and frequency of maximum power. These features capture the erratic, high-amplitude fluctuations characteristic of dyskinesia. In contrast, stiffness was best detected using both time- and frequency-domain features, including the maximum range and absolute mean differences, reflecting limited movement variability. The distribution of informative features reflects the physiological nature of each symptom and highlights the value of combining multiple sensors and feature types.

### 3.2. Overall State Evaluation

To predict the patient’s overall state, two experiments were conducted. The first one involved using data from a single sensor device—either the MYO armband or a smartphone—while the second experiment used data from both devices simultaneously. The metric values obtained through cross-validation are presented in Table 8. It showcases results received for predicting both the state according to the clinician and according to the patient.

In the case of patient state prediction, similar to symptom severity prediction, no significant differences were observed between the 10-fold (10F) and Leave-One-Out (LOO) validation splits. This suggests that the models are not prone to overfitting or data leakage, further confirming the robustness of the methodology. Interestingly, across all the prediction tasks, the best-performing models were consistently those based on SVMs, demonstrating their high effectiveness in handling this type of biomedical data. This was confirmed using the Wilcoxon signed-rank test: when comparing SVM results (*R*^2^) with those of RF and XG, *p*-values of 0.035 and 0.00024 were obtained, respectively (both lower than the significance level of 0.05).

As expected, models trained on combined data from both sensor devices outperformed those trained on data from a single device. This reinforces the necessity of using both sensors to collect comprehensive data. The improvement can be attributed to the fact that the sensors are positioned on different parts of the body during examination—the phone is held in the hand, while the MYO armband is worn on the forearm, thereby capturing a wider range of motion and providing complementary information.

The best models achieved strong predictive power, with correlations between true labels and predictions reaching values as high as 0.8. Notably, the model predicting the patient’s self-assessed state performed slightly better than the one based on the clinician’s evaluation. This was unexpected and may suggest that the patients’ own perceptions of their condition—when paired with sensor data—can be modeled more accurately.

To further analyze model performance, scatter plots are shown in Figure 7, illustrating the correlation between true values and predictions for both the patient’s and clinician’s assessments.

Both models demonstrate good predictive performance when estimating symptom severity (negative values), reflecting the motor deficits associated with PD. However, their ability to predict dyskinesias (positive values) is limited. Neither model was able to predict values higher than approximately 0.5, falling short of the upper range of possible scores (up to 4). This limitation is likely due to the small number of samples exhibiting pronounced dyskinesias in the dataset. This is consistent with the poor performance observed in the dyskinesia-specific model described earlier.

Overall, the results emphasize the importance of multimodal data collection and the strengths of SVMs in modeling patient states, while also highlighting the challenges of accurately capturing rarer symptoms such as dyskinesias.

To gain additional insights into the modeling process, feature importance was analyzed for the best overall state prediction models. This helped identify which features contributed most to the final predictions (Table 9).

The analysis revealed that both clinician- and patient-based models rely on sensor features reflecting movement variability and distribution. For clinician assessments, top features include absolute mean difference and interquartile range from accelerometer and gyroscope signals, along with the time since diagnosis. Patient models highlight similar features such as skewness, axis correlations, and the time since diagnosis. While slight differences appear in the specific metrics emphasized, these top features largely reflect aspects of motor function relevant to PD severity. This suggests that both perspectives capture overlapping information from sensor data.

## 4. Discussion and Conclusions

The experiments in this paper focused on building machine learning models to predict both the individual severities of symptoms and the overall state of patients related to PD, as assessed by clinicians and patients. Due to the limited and imbalanced dataset, shallow machine learning models were used.

The results reveal both the promise and the limitations of such models in clinical applications. The best performance was observed when data from all available exercises were combined, suggesting that aggregating diverse movement patterns improves model robustness. Tremor emerged as the most predictable symptom, likely due to its more visible and measurable nature in sensor data. In contrast, symptoms such as bradykinesia, stiffness, and dyskinesia were more difficult to assess. This may be attributed not only to their subtler manifestations but also to the significant class imbalance and the underrepresentation of higher severity levels in the dataset. These factors highlight a central challenge in PD symptom modeling: while wearable sensors can provide rich input, the clinical variability and skewed distribution of symptom severities can severely limit model generalizability and predictive power.

One notable finding was that models trained on patient self-assessments performed slightly better than those using clinician ratings. While this might initially seem surprising, it may reflect the fact that patients experience and recognize their symptoms throughout the day, while clinical evaluations are limited to short check-ups. This highlights the potential value of incorporating patient-reported data into monitoring systems, especially for symptoms that fluctuate. Another observation was that positive state values—indicating the presence and severity of dyskinesias on the TRS scale —were predicted less accurately, likely due to their underrepresentation in the dataset. Lastly, the best results were achieved when data from both sensors—the MYO armband and the smartphone—were used together, suggesting that combining multiple sources of information gives a fuller picture of symptom expression.

This study’s main limitations stem from the small dataset size and the imbalance in symptom severities, which likely hindered model performance for rarer or less obvious symptoms. Addressing this will require both broader data collection and possibly augmentation techniques to simulate underrepresented cases. Furthermore, although shallow models like SVMs and RF models were appropriate given the dataset, more complex models—such as deep neural networks or complex ensemble approaches—may uncover richer patterns if more data becomes available.

In conclusion, this work supports the feasibility of using wearable sensor data and machine learning to monitor PD symptoms, especially when models are tuned to specific symptoms and incorporate multimodal inputs. Future research should focus on expanding datasets, improving class balance, and exploring hybrid modeling approaches that blend patient and clinical insights. With these advancements, such models could become valuable tools for real-time, individualized disease monitoring and management in Parkinson’s disease.

## Figures and Tables

**Figure 1 sensors-25-04924-f001:**
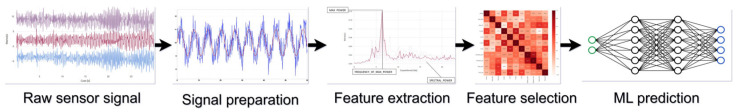
A chart presenting the preparation of the raw signal for conventional machine learning models.

**Figure 2 sensors-25-04924-f002:**
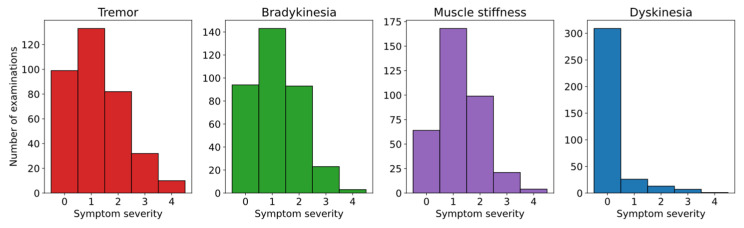
A histogram presenting the distribution of symptom severities for the dataset.

**Figure 3 sensors-25-04924-f003:**
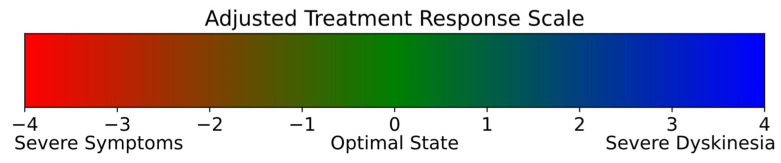
Value range of the adjusted TRS scale.

**Figure 4 sensors-25-04924-f004:**
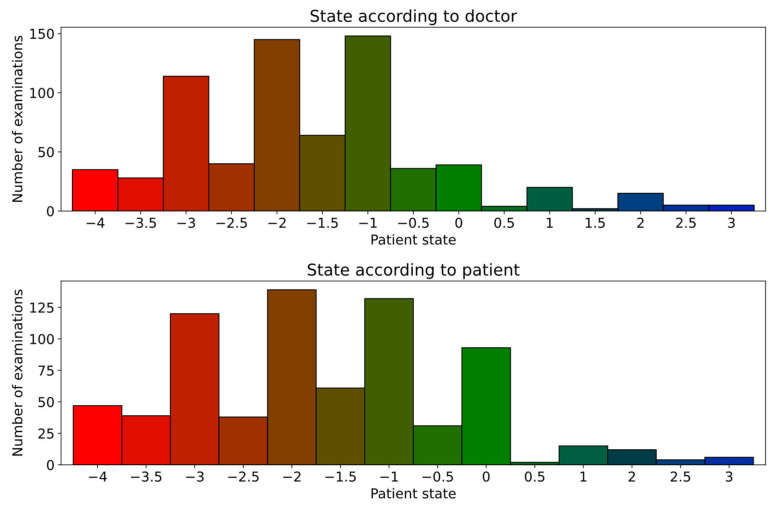
The distribution of label values representing the patient state evaluated by the clinician (**top**) and by the patient (**bottom**).

**Figure 5 sensors-25-04924-f005:**
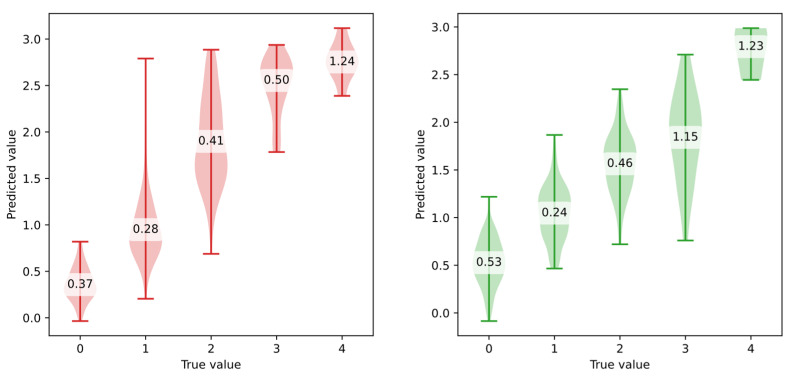
Violin plots presenting regression results with class-specific *MAE* values for tremor (**left**) and bradykinesia (**right**) using best-performing models evaluating based on a single exercise sensor signal.

**Figure 6 sensors-25-04924-f006:**
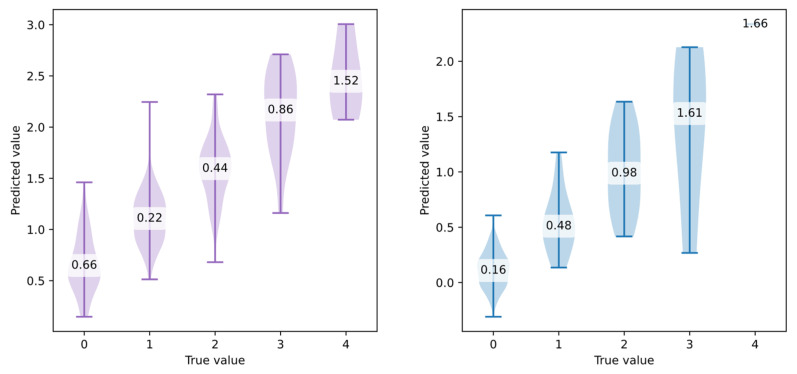
Violin plots presenting regression results with class-specific *MAE* values for muscle stiffness (**left**) and dyskinesia (**right**) using best-performing models evaluated based on a single exercise sensor signal.

**Figure 7 sensors-25-04924-f007:**
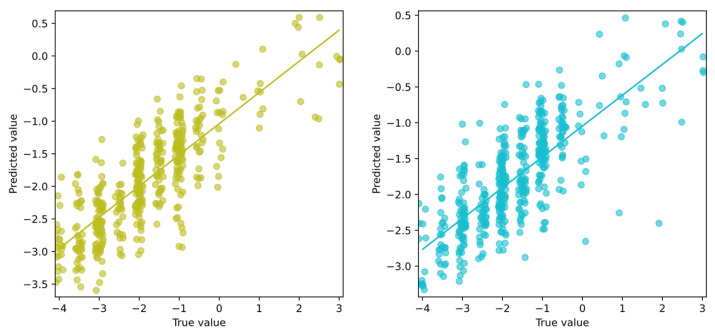
Scatter plots presenting regression results for predicting the patient’s overall state regarding PD using best-performing models trained on the patient’s self-evaluation (**left**) and clinician’s evaluations (**right**).

**Table 1 sensors-25-04924-t001:** Characteristics of the dataset.

Characteristic	Value
Total number of patients	241
Age (years) *	62.0 (11.1)
Years since diagnosis *	10.5 (6.10)
Patient sex	98 female, 143 male
Examination count	739
Examinations per patient *	3.07 (2.77)
States according to clinician *	−1.64 (1.38)
States according to patient *	−1.66 (1.42)
Examinations with state assessment according to doctor	700
Examinations with symptom assessment	356

*—represented by mean and standard deviation.

**Table 2 sensors-25-04924-t002:** List of features extracted from inertial sensor signals.

Feature	Equation/Explanation	
Time domain		
Mean	$\bar{x}=\frac{1}{n}\sum_{i=1}^{n}x_{i}$	(2)
Standard deviation	$s=\sqrt{\frac{1}{n-1}\sum_{i=1}^{n}{\left(x_{i}-\bar{x}\right)}^{2}}$	(3)
Median	The middle value of the sorted signal samples.	
Skewness	$S=\frac{n}{\left(n-1\right)\left(n-2\right)}\sum_{i=1}^{n}{\left(\frac{x_{i}-\bar{x}}{s}\right)}^{3}$	(4)
Kurtosis	$K=\frac{n\sum_{i=1}^{n}{\left(x_{i}-\bar{x}\right)}^{4}}{{\left(\sum_{i=1}^{n}{\left(x_{i}-\bar{x}\right)}^{2}\right)}^{2}}-3$	(5)
Max	The maximum value in the signal.	
Min	The minimum value in the signal.	
Interquartile range	The difference between the 75th and 25th percentiles of the signal.	
Approximate entropy	A measure of the regularity and unpredictability of fluctuations in a time series [28].	
Sample entropy	A measure of the likelihood that similar sequences in time-series data remain similar over time [28].	
Power	$P=\frac{1}{n}\sum_{i=1}^{n}x_{i}^{2}$	(6)
Absolute mean difference	$∆=\left|\frac{2}{n}\sum_{i=1}^{n/2}x_{i}-\frac{2}{n}\sum_{i=n/2+1}^{n}x_{i}\right|$	(7)
Frequency domain		
Max power	Maximum power found in the PSD.	
Max power frequency	The frequency at which the maximum power occurs.	
Spectral power	$P=\frac{1}{N}\sum_{i=0}^{N-1}{\left|X\left(f_{i}\right)\right|}^{2}$	(8)
Weighted mean power	$WMP=\frac{\sum_{i=0}^{N-1}{\left|X\left(f_{i}\right)\right|}^{2}\cdot f_{i}}{\sum_{i=0}^{N-1}f_{i}}$	(9)
Kurtosis	$K=\frac{N\sum_{i=1}^{N}{\left(\left|X\left(f_{i}\right)\right|-\bar{\left|X\left(f\right)\right|}\right)}^{4}}{{\left(\sum_{i=1}^{N}{\left(\left|X\left(f_{i}\right)\right|-\bar{\left|X\left(f\right)\right|}\right)}^{2}\right)}^{2}}-3$	(10)
Skewness	$S=\frac{N}{\left(N-1\right)\left(N-2\right)}\sum_{i=1}^{N}{\left(\frac{\left|X\left(f_{i}\right)\right|-\bar{\left|X\left(f\right)\right|}}{s}\right)}^{3}$	(11)
Interquartile range	Interquartile Range of the PSD values.	
Spectral centroid	$C=\frac{\sum_{i=0}^{N-1}f_{i}\cdot \left|X\left(f_{i}\right)\right|}{\sum_{i=0}^{N-1}\left|X\left(f_{i}\right)\right|}$	(12)

*n*—number of samples, *x_i_*—*i*-th sample, *N*—number of frequency bins, *f_i_*—frequency of the *i*-th bin, *X*(*f_i_*)—magnitude of the Fourier Transform at the *i*-th bin.

**Table 3 sensors-25-04924-t003:** Features created from patient and examination metadata.

Name	Description	Source
affected side	The side of the body more affected by the disease	patient
handedness	The dominant hand of the patient	patient
groups	Belonging to groups (disease, treatment method)	patient
diagnosis	Time since diagnosis to execution of examination	patient + exam
age	Age during examination	patient + exam

**Table 4 sensors-25-04924-t004:** Training results for models predicting symptom severities based on single sensor signals for single exercises.

Symptom	Split	Dataset	Model	*R* ^2^	*r*	*MAE*	*bMAE*	*bMSE*
bradykinesia	10F	Phone-ACC-#3	SVM	0.400	0.638	0.547	0.956	1.402
	LOO	MYO-GYRO-#3	XG	0.408	0.639	0.570	0.896	1.187
dyskinesia	10F	Phone-GYRO-#1	RF	0.385	0.622	0.228	1.238	2.353
	LOO	Phone-ACC-#1	SVM	0.355	0.641	0.240	1.379	2.837
stiffness	10F	MYO-ACC-#3	RF	0.309	0.562	0.568	0.992	1.402
	LOO	MYO-ACC-#3	XG	0.360	0.600	0.543	0.869	1.073
tremor	10F	Phone-ACC-#1	RF	0.595	0.772	0.537	0.675	0.652
	LOO	Phone-ACC-#1	XG	0.616	0.786	0.514	0.652	0.642

GYRO—gyroscope, ACC—accelerometer.

**Table 5 sensors-25-04924-t005:** Training results for models predicting symptom severities for single exercises.

Symptom	Split	Dataset	Model	*R* ^2^	*r*	*MAE*	*bMAE*	*bMSE*
bradykinesia	10F	#3	RF	0.422	0.654	0.556	0.956	1.412
	LOO	#3	SVM	0.392	0.631	0.562	0.977	1.437
dyskinesia	10F	#1	SVM	0.433	0.69	0.251	1.175	2.024
	LOO	#1	SVM	0.477	0.722	0.245	1.182	2.178
stiffness	10F	#3	SVM	0.402	0.64	0.511	0.882	1.152
	LOO	#3	SVM	0.439	0.672	0.503	0.860	1.085
tremor	10F	#1	RF	0.630	0.795	0.496	0.657	0.637
	LOO	#1	SVM	0.674	0.822	0.464	0.646	0.645

**Table 6 sensors-25-04924-t006:** Training results for models predicting symptom severities based on all data.

Symptom	Split	Model	*R* ^2^	*r*	*MAE*	*bMAE*	*bMSE*
bradykinesia	10F	SVM	0.632	0.826	0.438	0.722	0.762
	LOO	SVM	0.629	0.827	0.435	0.721	0.765
dyskinesia	10F	SVM	0.585	0.802	0.238	0.979	1.442
	LOO	SVM	0.567	0.790	0.245	0.983	1.432
stiffness	10F	SVM	0.604	0.817	0.420	0.750	0.842
	LOO	SVM	0.617	0.822	0.410	0.738	0.834
tremor	10F	SVM	0.777	0.888	0.382	0.576	0.526
	LOO	SVM	0.780	0.887	0.378	0.561	0.498

**Table 7 sensors-25-04924-t007:** Most relevant sensor-based features for symptom severity prediction.

Symptom	Device, Sensor, Exercise	Hand	Axis	Parameter	Score
Tremor	Phone-ACC-#1	Left	*Z*	Spectral centroid (0–25 Hz)	0.0269
	MYO-GYRO-#1	Right	*Z*	Weighted mean power (3–9 Hz)	0.0261
	MYO-GYRO-#1	Left	*Z*	Min of entropy (4 s window)	0.0242
	MYO-GYRO-#3	Left	*X*	Skewness of entropy (4 s window)	0.0238
	Phone-ACC-#1	Right	*M*	Kurtosis (3–9 Hz)	0.0230
Bradykinesia	Phone-ACC-#3	Right	*M*	Skewness of value range (4 s window)	0.0406
	Phone-ACC-#3	Left	*X*	Mean of entropy (4 s window)	0.0371
	MYO-ACC-#1	Right	*X*	Median	0.0342
	MYO-ACC-#3	Left	*Y*	Max power (0–25 Hz)	0.0335
	Phone-ACC-#3	Left	*Z*	Absolute mean difference	0.0329
Dyskinesia	Phone-GYRO-#1	Left	*Z*	Spectral power	0.0694
	MYO-GYRO-#3	Right	*Z*	Interquartile range	0.0615
	MYO-ACC-#1	Left	*Y*	Frequency of max power	0.0458
	Phone-GYRO-#1	Left	*Z*	D1	0.0434
	Phone-GYRO-#3	Right	*Y*	Mean PSD	0.0410
Stiffness	Phone-GYRO-#3	Right	*Z*	Max of value range (4 s window)	0.0685
	Phone-ACC-#1	Left	*Y*	Absolute mean difference	0.0504
	Phone-ACC-#3	Left	*Y*	Mean PSD (9–14 Hz)	0.0489
	MYO-ACC-#3	Left	*Z*	Skewness	0.0482
	Phone-GYRO-#1	Left	*Y*	Min of entropy (4 s window)	0.0447

**Table 8 sensors-25-04924-t008:** Training results for models predicting patient’s state.

State According to	Split	Dataset	Model	*R* ^2^	*r*	*MAE*	*bMAE*	*bMSE*
clinician	10F	All	SVM	0.543	0.767	0.617	1.309	2.786
	LOO		SVM	0.534	0.758	0.629	1.309	2.765
	10F	MYO	SVM	0.468	0.703	0.67	1.413	3.209
	LOO		SVM	0.471	0.704	0.675	1.397	3.122
	10F	Phone	SVM	0.406	0.652	0.695	1.514	3.716
	LOO		SVM	0.409	0.653	0.697	1.505	3.693
patient	10F	All	SVM	0.610	0.816	0.603	1.144	2.232
	LOO		SVM	0.608	0.812	0.610	1.131	2.155
	10F	MYO	SVM	0.454	0.689	0.723	1.311	2.907
	LOO		SVM	0.452	0.687	0.724	1.306	2.878
	10F	Phone	SVM	0.436	0.674	0.729	1.378	3.166
	LOO		SVM	0.396	0.639	0.760	1.433	3.384

**Table 9 sensors-25-04924-t009:** Most relevant features for overall state prediction.

According To	Device, Sensor, Exercise	Hand	Axis	Parameter	Score
clinician	Phone-ACC-#1	Left	*Z*	Absolute mean difference	0.0365
	Phone-GYRO-#1	Left	*X*	Interquartile range (0–3 Hz)	0.0313
	MYO-GYRO-#3	Left	*Z*	Maximum	0.0307
	-	-	-	Time since diagnosis	0.0285
	Phone-GYRO-#1	Right	*X*	Weighted mean power (0–3 Hz)	0.0231
patient	MYO-ACC-#1	Left	*Y*	Skewness (0–25 Hz)	0.0329
	MYO-ACC-#3	Left	-	Correlation (*Y* and *Z*)	0.0262
	-	-	-	Time since diagnosis	0.0231
	MYO-GYRO-#1	Right	-	Correlation (*X* and *Y*)	0.0222
	Phone-ACC-#1	Left	*Z*	Spectral centroid (0–25 Hz)	0.0217

## Data Availability

The original contributions presented in the study are included in the article. Further inquiries can be directed to the corresponding author.

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
