# Peer review of "Machine Learning-Based Assessment of Parkinson’s Disease Symptoms Using Wearable and Smartphone Sensors"

_sensors, 2025, doi:10.3390/s25164924_

Round 1
Reviewer 1 Report
Comments and Suggestions for Authors
I'm not an expert in the mdeical domain, so my review is from the maching learning point of view.
The paper is overall well-written and has a potential, however, there are some limitations that have to be addressed:
1. The claim "The dataset contains a limited number of samples, therefore the deep learning methods that are nowadays often used did not yield significant results." is arguable. Nowadays machine learning technologies provide various sophisticated augmentation techniques. Besides, some pre-trained on other medical applications models can be tried.
2. Some more recent references could be beneficial if they exist in the PD or related domains (so, this is rather an optional improvement suggestions).
3. The feature selection procedure is not justified. Why not to use e.g. permutation importance instead?
4. The Multilayer Perceptron model is introduced in sec. 2.6 but then it is just forgotten. MLP is a common name for multilayer fully connected neural netoworks, and it is completely unclear what "default parameters" are used. Besides, modern transformer-based neural networks may perform fine on limited datasets.
5. There are some broken references in lines 461-462.
Some other suggestions:
1. Please, state clearly the scientific novelty and contribution of your work (in the introdiction).
2. I would suggest to avoid the term "traditional ML models" since many neural network-based models today can also be considered as "traditional ML models".
Author Response
We sincerely appreciate the reviewer’s thorough and insightful comments, which have helped us improve the quality and clarity of our manuscript. We have carefully considered each suggestion and incorporated changes to address the points raised. Here are our detailed responses:
Comments 1: The claim "The dataset contains a limited number of samples, therefore the deep learning methods that are nowadays often used did not yield significant results." is arguable. Nowadays machine learning technologies provide various sophisticated augmentation techniques. Besides, some pre-trained on other medical applications models can be tried.
Response 1: This claim is based on our conducted experiments. In a previous paper (Gutowski, T. Deep Learning for Parkinson’s Disease Symptom Detection and Severity Evaluation Using Accelerometer Signal, 2022, pp. 271–276, doi:10.14428/ESANN/2022.ES2022-107) and a related PhD thesis (https://bip.wat.edu.pl/bip/dokumenty/postepowania-awansowe/tgutowski/1_rozprawa_dr_tomasz_gutowski.pdf), a deep learning model was developed to predict the intensity and presence of symptoms (bradykinesia, tremor, and dyskinesia) using a significantly larger dataset funded by the Michael J. Fox Foundation (MJFF). These models performed well on the MJFF dataset. However, when the same models were trained on the data collected in the current study—including attempts using transfer learning with pre-trained models—the results were significantly worse than those presented here, which is why these experiments were excluded from the manuscript. Furthermore, as shown in Chapter 4.1.3 of the thesis, the MJFF dataset was also used to train conventional machine learning models, which achieved results similar to those of the deep learning methods (Tables 10 and 15 in the thesis).
Comments 2: Some more recent references could be beneficial if they exist in the PD or related domains (so, this is rather an optional improvement suggestions).
Response 2: We have incorporated recent references from 2024 and 2025 in the revised manuscript.
Comment 3: The feature selection procedure is not justified. Why not to use e.g. permutation importance instead?
Response 3: Our original approach involves two steps: first, removing highly correlated features (Pearson’s correlation > 0.97) to reduce redundancy, and second, an iterative, performance-driven feature selection during model training, where features are added based on improvement in cross-validated predictive accuracy. This ensures that selected features contribute directly to model generalization. Permutation importance has known limitations that are relevant in considering our study. When features are correlated, permuting one feature does not fully remove the related information since correlated features remain, leading to underestimated importance values. Also, permutation importance evaluates features relative to a fixed model, which can be unstable and less reliable with high-dimensional, small-sample datasets like ours. (https://scikit-learn.org/stable/modules/permutation_importance.html) Based on this suggestion we conducted some experiments with permutation importance and we decided to use it to evaluate the features used by the models, since at that point the number of features selected is lower and therefore less chance for higher correlations. To see which features are most important considering specific symptoms.
Comments 4: The Multilayer Perceptron model is introduced in sec. 2.6 but then it is just forgotten. MLP is a common name for multilayer fully connected neural netoworks, and it is completely unclear what "default parameters" are used. Besides, modern transformer-based neural networks may perform fine on limited datasets.
Response 4: MLP was initially considered, which is why it was included in the Materials and Methods section. However, during training, it became clear that MLP did not yield the best results in any experiment. Therefore, it was removed from the manuscript. Unfortunately, one mention remained, which has now been fully removed for clarity.
Comments 5: There are some broken references in lines 461-462.
Response 5: Thank you, these are now fixed.
Comments 6: Please, state clearly the scientific novelty and contribution of your work (in the introdiction).
Response 6: The introduction has been updated to state the novelty and contribution more clearly.
Comments 7: I would suggest to avoid the term "traditional ML models" since many neural network-based models today can also be considered as "traditional ML models".
Response 7: All instances of this term have been replaced with “shallow ML models” throughout the manuscript.
Reviewer 2 Report
Comments and Suggestions for Authors
The manuscript addresses an important and timely topic: the objective assessment of Parkinson's disease (PD) symptom severity using machine learning (ML) on data from wearable and smartphone sensors. The study leverages multimodal sensor data and a variety of ML models to predict both individual symptom severities and overall patient state. The structure is generally clear, and the results show practical potential for personalized medicine and remote monitoring. However, several significant areas should be addressed before the study is considered for publication.
1. While the authors acknowledge the small and imbalanced dataset, the limitations this poses are not fully addressed. Consider implementing data augmentation techniques, re-sampling, or using robust statistical approaches to partially mitigate these issues. Additionally, discuss the possible impact of class imbalance on the clinical utility of the developed models.
Please expand the discussion on how generalizable the findings are, given the current dataset size and representativeness.
2. The manuscript states that default parameters were used for all models. For a fair and strong comparison, models should be thoroughly tuned (e.g., via grid search, cross-validated hyperparameter optimization). Please clarify if any such process was carried out and, if not, consider adding it.
3. In the current form, the rationale for selecting SVM/RF/XGBoost/MLP is insufficient. Was model performance statistically compared? Consider providing statistical tests to support your model ranking.
4. The manuscript provides an extensive feature extraction pipeline but lacks discussion regarding feature interpretability and its potential clinical relevance. Are there particular features most predictive of certain symptoms? If so, highlight these and discuss their physiological plausibility.
5. Please provide at least a brief feature importance analysis, especially since clinical deployment depends on transparent models.
Clinical Ground Truth and Labeling
6. There seems to be limited explanation for potential sources of inconsistency between clinician and patient assessments. Please elaborate on how these two sources of ground truth were handled in the models, and discuss the reliability of each.
The substantial difference observed between patient- and clinician-based predictions should be discussed in more depth, possibly referencing related literature.
7. Some predicted values are outside the valid range (e.g., negative values for symptom severity). Please discuss methods for constraining outputs, such as bounding predictions within valid ranges.
8. Update the literature review to include recent studies and provide a comprehensive overview of the field, (i)Ye Y S, Chen M R, Zou H L, et al. GID: Global information distillation for medical semantic segmentation[J]. Neurocomputing. (ii) hang Q, Huang A, Shao L, et al. A machine learning framework for identifying influenza pneumonia from bacterial pneumonia for medical decision making[J]. Journal of Computational Science. (iii) Li X, Fu Q, Li Q, et al. Multi-objective binary grey wolf optimization for feature selection based on guided mutation strategy. Applied Soft Computing. (iv) Qian L, Huang H, Xia X, et al. Automatic segmentation method using FCN with multi-scale dilated convolution for medical ultrasound image. The Visual Computer. Please ensure to cite.
9. Please elaborate on clinical implications of the current MAE/bMAE results. For instance, what would be an acceptable error margin from a clinical perspective?
Author Response
We appreciate Reviewer 2’s careful evaluation and valuable feedback. Their insightful comments have helped us improve the clarity and robustness of our work. Below, we provide the reviewer’s comments followed by our detailed responses.
Comments: 1: While the authors acknowledge the small and imbalanced dataset, the limitations this poses are not fully addressed. Consider implementing data augmentation techniques, re-sampling, or using robust statistical approaches to partially mitigate these issues. Additionally, discuss the possible impact of class imbalance on the clinical utility of the developed models. Please expand the discussion on how generalizable the findings are, given the current dataset size and representativeness.
Response 1: We previously conducted experiments involving data augmentation and re-sampling. However, evaluation metrics such as MAE, MSE, R², and correlation are highly dependent on the distribution of severity levels in the data. When the distributions between training and test sets differ significantly, test performance may appear poor even if the model learns relevant patterns effectively. Maintaining comparable distributions between training and test sets is therefore essential. Given the small number of samples in some severity classes (e.g., only one sample for dyskinesia severity 4), it is not feasible to augment only the training set without distorting this balance. Including synthetic data in the test set, on the other hand, would reduce the clinical validity of the evaluation. While this limits the performance of our models, we believe that using only real patient data for both training and evaluation ensures a valid and transparent assessment of model performance under realistic clinical conditions. Our future studies would benefit from larger, more balanced datasets to further improve performance and applicability.
Comments 2: The manuscript states that default parameters were used for all models. For a fair and strong comparison, models should be thoroughly tuned (e.g., via grid search, cross-validated hyperparameter optimization). Please clarify if any such process was carried out and, if not, consider adding it.
Response 2: In this study, we used default model parameters primarily to establish a consistent and interpretable baseline across all models. Initial experiments involving limited hyperparameter adjustments indicated only marginal improvements in performance, suggesting that default settings already provided reasonably strong results. In a way we have tuned the models through systematic selection of optimal feature subsets, testing multiple configurations including different datasets, models, data splits, and sensor signal combinations. As detailed in tables in the manuscript and Tables A1 and A2, over 800 distinct training configurations were evaluated to identify the best-performing models. Introducing a full grid search over even a small hyperparameter space (e.g., 4 parameters with 4 values each) combined with cross-validation and leave-one-patient-out splits would vastly increase the number of training runs—potentially exceeding 3 million—making it computationally difficult with the current resources. Thus, our approach balances thorough exploration of model configurations with practical constraints.
Comments 3: In the current form, the rationale for selecting SVM/RF/XGBoost/MLP is insufficient. Was model performance statistically compared? Consider providing statistical tests to support your model ranking.
Response 3: The rationale for selecting SVM, RF, XGBoost, and MLP has been clarified, and statistical comparisons of model performance have been included. We updated the manuscript: “Interestingly, across all the prediction tasks, the best-performing models were consistently those based on SVMs, demonstrating their high effectiveness in handling this type of biomedical data. This was confirmed using the Wilcoxon signed-rank test: when comparing SVM results (R2) those of RF and XG, p-values of 0.035 and 0.00024 were obtained, respectively (both lower than significance level of 0.05).”
Comments 4: The manuscript provides an extensive feature extraction pipeline but lacks discussion regarding feature interpretability and its potential clinical relevance. Are there particular features most predictive of certain symptoms? If so, highlight these and discuss their physiological plausibility.
Response 4: Additional experiments were conducted and using permutation importance (as suggested by another reviewer) scores were calculated for features. This allowed to identify what characteristics of movement were most important in symptom and overall state evaluation.
Comments 5: Please provide at least a brief feature importance analysis, especially since clinical deployment depends on transparent models.
Response 5: As stated in an answer to comment #4 brief feature analysis was added to the manuscript.
Comments 6: There seems to be limited explanation for potential sources of inconsistency between clinician and patient assessments. Please elaborate on how these two sources of ground truth were handled in the models, and discuss the reliability of each. The substantial difference observed between patient- and clinician-based predictions should be discussed in more depth, possibly referencing related literature.
Response 6: A detailed comparison between patient and clinician ratings is beyond the scope of this study. However, we have added a paragraph discussing potential reasons for differences in their evaluations: "Clinician assessments tend to reflect standardized diagnostic criteria and are influenced by medical training and experience, whereas patient self-reports are subjective and may be shaped by individual perceptions or mood. Previous research [35,36] has also highlighted discrepancies between patient- and clinician-reported outcomes, especially in conditions involving fluctuating or non-visible symptoms of the disease. "
Comments 7: Some predicted values are outside the valid range (e.g., negative values for symptom severity). Please discuss methods for constraining outputs, such as bounding predictions within valid ranges.
Response 7: In the manuscript we added 3 proposed methods and selected one this is best in this scenario:
1. Using post-processing to clip predicted values to the valid range [0, 4],
2. Reformulating the problem as a classification task (ordinal classification). However, this would lead to a loss of prediction precision.
3. If neural network were explored, applying a bounded activation function scaled to the target range in the final layer of the model.
Considering the advantages and disadvantages of these methods, clipping the values to the valid range is the best solution for this problem.
Comments 8: Update the literature review to include recent studies and provide a comprehensive overview of the field, (i)Ye Y S, Chen M R, Zou H L, et al. GID: Global information distillation for medical semantic segmentation[J]. Neurocomputing. (ii) hang Q, Huang A, Shao L, et al. A machine learning framework for identifying influenza pneumonia from bacterial pneumonia for medical decision making[J]. Journal of Computational Science. (iii) Li X, Fu Q, Li Q, et al. Multi-objective binary grey wolf optimization for feature selection based on guided mutation strategy. Applied Soft Computing. (iv) Qian L, Huang H, Xia X, et al. Automatic segmentation method using FCN with multi-scale dilated convolution for medical ultrasound image. The Visual Computer. Please ensure to cite.
Response 8: Studies from 2024 and 2025 have been added to the manuscript, and the suggested paper by Li et al. (Multi-objective binary grey wolf optimization for feature selection based on guided mutation strategy, Applied Soft Computing) has been cited accordingly. However, the other referenced papers, while valuable, are not closely related to the specific focus of our study and were therefore not included.
Comments 9: . Please elaborate on clinical implications of the current MAE/bMAE results. For instance, what would be an acceptable error margin from a clinical perspective?
Response 9: Clinicians typically evaluate Parkinson’s symptoms using discrete severity scores ranging from 0 to 4 (both in this study and in the MDS-UPDRS). An MAE or bMAE value below 1, therefore, indicates that the model’s predictions, on average, deviate by less than one severity level. This is clinically meaningful and suggests a level of precision that approaches—or even exceeds—that of standard clinical assessments. However, it is important to note that these metrics represent averages across all samples; some predictions may have small errors, while others may exceed one severity level. When examining class-specific MAE values (bMAE), we observe that for tremor and stiffness, an error greater than 1 occurs only at the highest severity level (4). For bradykinesia, it is exceeded at severity levels 3 and 4. This suggests that the models perform adequately for lower severity levels but require further refinement—and likely more training data—to improve accuracy in detecting the most severe cases. Nevertheless, this level of accuracy is promising, as it implies that the model could reliably support clinical decision-making by providing symptom severity estimates that are at least as consistent as those from clinician ratings.
Reviewer 3 Report
Comments and Suggestions for Authors
The topic is timely and relevant, and the application of machine learning to wearable sensor data for Parkinson’s disease assessment represents a valuable contribution to the field. The manuscript is generally well-structured and presents a comprehensive pipeline from data acquisition to model evaluation.
With careful attention to the issues noted below—particularly grammatical accuracy, clarification of dataset details, and completeness of methodological descriptions—the clarity and accessibility of the manuscript can be significantly improved.
Below are my detailed comments:
1. Introduction
(Line 31) Please add the definite article “the”:
“are most widely recognized features of PD” → “are the most widely recognized features of PD”
(Line 45) Please insert the missing conjunction:
“…sleep problems, and constipation.”
(Line 49) Please correct the spelling error:
“neurotransmier” → “neurotransmitter”
(Line 82) Please remove the comma after “both” for grammatical accuracy.
(Line 91) Please add a comma after “Furthermore” to improve readability:
2. Materials and Methods
(Line 104) Please remove the comma before “and” for correct punctuation in a compound subject.
(Line 108) Please correct the spelling:
"...which has been approved by the Bioethics Commiee..." → "...approved by the Bioethics Committee..."
(Line 133) Please consider rephrasing for fluency:
"This is followed by the pronation-supination task performed also for 30 seconds..." → "Next, the patient performs a 30-second pronation-supination task..."
(Line 137) Please add a comma for clarity:
"After each examination, performed under clinical supervision the clinician evaluates..." → "After each examination, performed under clinical supervision, the clinician evaluates..."
(Lines 139-142 ) There is a subject–verb agreement issue: "provide" should be "provides", or the sentence should be rephrased for clarity.
(Lines 148-151) The sentence is too long and lacks clarity. Consider splitting and rephrasing it to improve readability.
(Line 152) The text mentions that the dataset includes 352 PD patients, while Table 1 reports 241 patients. If 352 refers to the broader dataset and 241 is the final subset used in the analysis, please clarify this distinction to avoid confusion.
(Line 214) Please revise the punctuation for consistency in listing:
“...SciPy and PyWavelets, and EntropyHub Python libraries.” → “...SciPy, PyWavelets, and EntropyHub Python libraries.”
(Line 260) Please correct the usage and pluralization:
"Having duplicates features..." → "Having duplicate features..."
(Line 303) Please correct the spelling:
"...might now give enough information..." → "...might not provide sufficient information..."
(Lines 282-296) The section lists the Multilayer Perceptron (MLP) model but does not provide any description. A brief 1–2 sentence explanation of MLP and its application in regression should be added for consistency with the other models.
4. Discussion and Conclusions
- The phrase "positive state values (experiencing dyskinesias)" may be unclear to readers unfamiliar with the TRS scale. Consider briefly clarifying that, in this context, positive values indicate the presence of dyskinesia, which differs from typical clinical scoring systems where higher values generally correspond to greater symptom severity.
The manuscript is generally well-written and clear. However, there are a few grammatical errors, typographical issues, and awkward phrasings throughout the text. A careful proofreading would improve the overall readability and polish the language.
Author Response
We thank you for your valuable suggestions, which have helped us improve the manuscript’s clarity and accuracy. All the grammatical corrections and stylistic improvements have been incorporated accordingly.
Regarding the dataset clarification, we confirm that the final analysis included data from 241 Parkinson’s disease patients. The larger number of 352 patients refers to the dataset, which also contained individuals with related conditions such as essential tremor and atypical parkinsonisms.
Concerning the Multilayer Perceptron (MLP) model, it has been fully removed from the manuscript, as it did not outperform other methods during our experiments.
Reviewer 4 Report
Comments and Suggestions for Authors The study confirms the viability of using data from wearable sensors and machine learning models to monitor Parkinson's disease (PD) symptoms, suggesting real potential for personalized and real-time monitoring. The best results were obtained by combining data from multiple movement types, indicating that aggregating the diversity of motor patterns increases the robustness of the model. Models trained on patient self-reports provided better results than those based on clinical assessments, suggesting the value of subjective data in complementing objective ones, especially for fluctuating symptoms. The best performances were obtained by integrating data from the MYO armband and smartphone, demonstrating the benefits of multimodal input in understanding the complexity of PD symptoms. As aspects that can be improved: - the dataset was small, which limits the models' ability to generalize and reduces confidence in the results for rarer or more complex cases. - there is an underrepresentation of high levels of severity for some symptoms (e.g. dyskinesia), affecting the accuracy and learning capacity of the models. - symptoms such as rigidity and dyskinesia are more subtle and more difficult to capture by sensors, which reduces the accuracy of predictions for these conditions. - only classical models (SVM, regressions) were used, which may be insufficient to detect complex patterns in data with high variability. More sophisticated models (e.g. neural networks) were not explored due to data restrictions. - clinical assessments, although standardized, are limited to point-in-time, not capturing daily fluctuations in symptoms, which may introduce bias in model training. - the best performances were only obtained by the combined use of MYO and smartphone, which could limit scalability and practical applicability in real-world environments where not all equipment is available.Author Response
Thank you very much for your thoughtful summary and insightful observations regarding our study. We appreciate your recognition of the potential of wearable sensor data combined with machine learning to monitor Parkinson’s disease symptoms and the value of multimodal data integration. We also acknowledge the limitations you highlighted, including the small dataset size, underrepresentation of severe symptom levels, and challenges in capturing subtle symptoms like stiffness and dyskinesia. Your points regarding the constraints of classical machine learning models and the limitations of clinical assessments in reflecting symptom fluctuations are considered. We have used your review to improve the manuscript and clarify these important aspects.
Reviewer 5 Report
Comments and Suggestions for Authors
First, my congratulation to excellent results and their very clear presentation. I have only two small remarks:
- In comments to figures 5 and 6, you mentioned that level 0 was not always predicted (stiffness and bradykinesia). Did you try to find out the possible reason?
- I noticed that there are no references from last 2 years (2024 and 2025). Of course, there are many studies focused on gait analysis in relation to PD. Can you compare those approaches with yours, focused on upper limb?
There are only few minor issues. I would recommend revision by a native speaker or a good software tool.
Author Response
We thank you for your feedback and insightful comment. Below, we address the comments:
Comments 1: In comments to figures 5 and 6, you mentioned that level 0 was not always predicted (stiffness and bradykinesia). Did you try to find out the possible reason?
Response 1: The class distribution for stiffness and bradykinesia shows that the level 1 has the most samples. Therefore, the model prefers to make the predictions closer to 1 than 0 for ambigiuous examinations. These ambigiuous examinations could be the result of the fact that the examinations were performed by different clinicians therefore, leading to between raters variability. Furthermore, severity level 0 in these two symptoms often corresponds to extended periods of immobility, making it harder for the models to distinguish from low but nonzero symptom levels. The manuscript has been updated with the explanation.
Comments 2: I noticed that there are no references from last 2 years (2024 and 2025). Of course, there are many studies focused on gait analysis in relation to PD. Can you compare those approaches with yours, focused on upper limb?
Response 2: We have added more recent references from 2024 and 2025 related to similar studies. However, since our study focuses exclusively on upper limb movement and does not analyze gait, direct comparison with gait-focused studies is challenging. We referenced studies that conducted similar experiments, such as Thomas et al. (2018), which examined treatment-response indices from wearable sensors in PD. Comparing results across studies with different settings, datasets, and predicted outcomes risks confusing the factors responsible for performance differences, making meaningful comparison difficult.
Comments 3: There are only few minor issues. I would recommend revision by a native speaker or a good software tool.
Response 3: The manuscript has been revised to fix these issues.
Round 2
Reviewer 1 Report
Comments and Suggestions for Authors
The authors has done a good job improving the paper. However, two of the comments (1 and 3) have been extencively eplained in the answers to the reviewer, but not in the paper. I consider the explanations important for justification of researh decisions, so they should be included in the paper . Maybe in a shorter form, but preserving references (except one to scikit-learn).
Author Response
Thank you very much for your valuable feedback. We have now incorporated explanations addressing Comments 1 and 3 directly into the manuscript. The explanation regarding Comment 1 can be found in lines 160–169, and for Comment 3 in lines 282–290. We appreciate your helpful guidance.
Reviewer 2 Report
Comments and Suggestions for Authors
The manuscript addresses an important and timely topic: the objective assessment of Parkinson's disease (PD) symptom severity using machine learning (ML) on data from wearable and smartphone sensors. The study leverages multimodal sensor data and a variety of ML models to predict both individual symptom severities and overall patient state. The structure is generally clear, and the results show practical potential for personalized medicine and remote monitoring. However, several significant areas should be addressed before the study is considered for publication.
1. While the authors acknowledge the small and imbalanced dataset, the limitations this poses are not fully addressed. Consider implementing data augmentation techniques, re-sampling, or using robust statistical approaches to partially mitigate these issues. Additionally, discuss the possible impact of class imbalance on the clinical utility of the developed models.
Please expand the discussion on how generalizable the findings are, given the current dataset size and representativeness.
2. The manuscript states that default parameters were used for all models. For a fair and strong comparison, models should be thoroughly tuned (e.g., via grid search, cross-validated hyperparameter optimization). Please clarify if any such process was carried out and, if not, consider adding it.
3. In the current form, the rationale for selecting SVM/RF/XGBoost/MLP is insufficient. Was model performance statistically compared? Consider providing statistical tests to support your model ranking.
4. The manuscript provides an extensive feature extraction pipeline but lacks discussion regarding feature interpretability and its potential clinical relevance. Are there particular features most predictive of certain symptoms? If so, highlight these and discuss their physiological plausibility.
5. Please provide at least a brief feature importance analysis, especially since clinical deployment depends on transparent models.
Clinical Ground Truth and Labeling
6. There seems to be limited explanation for potential sources of inconsistency between clinician and patient assessments. Please elaborate on how these two sources of ground truth were handled in the models, and discuss the reliability of each.
The substantial difference observed between patient- and clinician-based predictions should be discussed in more depth, possibly referencing related literature.
7. Some predicted values are outside the valid range (e.g., negative values for symptom severity). Please discuss methods for constraining outputs, such as bounding predictions within valid ranges.
8. Update the literature review to include recent studies and provide a comprehensive overview of the field, (i)Ye Y S, Chen M R, Zou H L, et al. GID: Global information distillation for medical semantic segmentation[J]. Neurocomputing. (ii) hang Q, Huang A, Shao L, et al. A machine learning framework for identifying influenza pneumonia from bacterial pneumonia for medical decision making[J]. Journal of Computational Science. (iii) Li X, Fu Q, Li Q, et al. Multi-objective binary grey wolf optimization for feature selection based on guided mutation strategy. Applied Soft Computing. (iv) Qian L, Huang H, Xia X, et al. Automatic segmentation method using FCN with multi-scale dilated convolution for medical ultrasound image. The Visual Computer. Please ensure to cite.
9. Please elaborate on clinical implications of the current MAE/bMAE results. For instance, what would be an acceptable error margin from a clinical perspective?
Author Response
Dear Reviewer,
Thank you for your continued evaluation of our manuscript.
We noticed that your second-round review is exactly the same as in the first round, without addressing the revisions we made or providing any new feedback.
We have carefully addressed all your original comments and provided detailed responses and manuscript revisions.
Below, please find our detailed responses to the comments you submitted:
Comments: 1: While the authors acknowledge the small and imbalanced dataset, the limitations this poses are not fully addressed. Consider implementing data augmentation techniques, re-sampling, or using robust statistical approaches to partially mitigate these issues. Additionally, discuss the possible impact of class imbalance on the clinical utility of the developed models. Please expand the discussion on how generalizable the findings are, given the current dataset size and representativeness.
Response 1: We previously conducted experiments involving data augmentation and re-sampling. However, evaluation metrics such as MAE, MSE, R², and correlation are highly dependent on the distribution of severity levels in the data. When the distributions between training and test sets differ significantly, test performance may appear poor even if the model learns relevant patterns effectively. Maintaining comparable distributions between training and test sets is therefore essential. Given the small number of samples in some severity classes (e.g., only one sample for dyskinesia severity 4), it is not feasible to augment only the training set without distorting this balance. Including synthetic data in the test set, on the other hand, would reduce the clinical validity of the evaluation. While this limits the performance of our models, we believe that using only real patient data for both training and evaluation ensures a valid and transparent assessment of model performance under realistic clinical conditions. Our future studies would benefit from larger, more balanced datasets to further improve performance and applicability.
Comments 2: The manuscript states that default parameters were used for all models. For a fair and strong comparison, models should be thoroughly tuned (e.g., via grid search, cross-validated hyperparameter optimization). Please clarify if any such process was carried out and, if not, consider adding it.
Response 2: In this study, we used default model parameters primarily to establish a consistent and interpretable baseline across all models. Initial experiments involving limited hyperparameter adjustments indicated only marginal improvements in performance, suggesting that default settings already provided reasonably strong results. In a way we have tuned the models through systematic selection of optimal feature subsets, testing multiple configurations including different datasets, models, data splits, and sensor signal combinations. As detailed in tables in the manuscript and Tables A1 and A2, over 800 distinct training configurations were evaluated to identify the best-performing models. Introducing a full grid search over even a small hyperparameter space (e.g., 4 parameters with 4 values each) combined with cross-validation and leave-one-patient-out splits would vastly increase the number of training runs—potentially exceeding 3 million—making it computationally difficult with the current resources. Thus, our approach balances thorough exploration of model configurations with practical constraints.
Comments 3: In the current form, the rationale for selecting SVM/RF/XGBoost/MLP is insufficient. Was model performance statistically compared? Consider providing statistical tests to support your model ranking.
Response 3: The rationale for selecting SVM, RF, XGBoost, and MLP has been clarified, and statistical comparisons of model performance have been included. We updated the manuscript: “Interestingly, across all the prediction tasks, the best-performing models were consistently those based on SVMs, demonstrating their high effectiveness in handling this type of biomedical data. This was confirmed using the Wilcoxon signed-rank test: when comparing SVM results (R2) those of RF and XG, p-values of 0.035 and 0.00024 were obtained, respectively (both lower than significance level of 0.05).”
Comments 4: The manuscript provides an extensive feature extraction pipeline but lacks discussion regarding feature interpretability and its potential clinical relevance. Are there particular features most predictive of certain symptoms? If so, highlight these and discuss their physiological plausibility.
Response 4: Additional experiments were conducted and using permutation importance (as suggested by another reviewer) scores were calculated for features. This allowed to identify what characteristics of movement were most important in symptom and overall state evaluation.
Comments 5: Please provide at least a brief feature importance analysis, especially since clinical deployment depends on transparent models.
Response 5: As stated in an answer to comment #4 brief feature analysis was added to the manuscript.
Comments 6: There seems to be limited explanation for potential sources of inconsistency between clinician and patient assessments. Please elaborate on how these two sources of ground truth were handled in the models, and discuss the reliability of each. The substantial difference observed between patient- and clinician-based predictions should be discussed in more depth, possibly referencing related literature.
Response 6: A detailed comparison between patient and clinician ratings is beyond the scope of this study. However, we have added a paragraph discussing potential reasons for differences in their evaluations: "Clinician assessments tend to reflect standardized diagnostic criteria and are influenced by medical training and experience, whereas patient self-reports are subjective and may be shaped by individual perceptions or mood. Previous research [35,36] has also highlighted discrepancies between patient- and clinician-reported outcomes, especially in conditions involving fluctuating or non-visible symptoms of the disease. "
Comments 7: Some predicted values are outside the valid range (e.g., negative values for symptom severity). Please discuss methods for constraining outputs, such as bounding predictions within valid ranges.
Response 7: In the manuscript we added 3 proposed methods and selected one this is best in this scenario:
1. Using post-processing to clip predicted values to the valid range [0, 4],
2. Reformulating the problem as a classification task (ordinal classification). However, this would lead to a loss of prediction precision.
3. If neural network were explored, applying a bounded activation function scaled to the target range in the final layer of the model.
Considering the advantages and disadvantages of these methods, clipping the values to the valid range is the best solution for this problem.
Comments 8: Update the literature review to include recent studies and provide a comprehensive overview of the field, (i)Ye Y S, Chen M R, Zou H L, et al. GID: Global information distillation for medical semantic segmentation[J]. Neurocomputing. (ii) hang Q, Huang A, Shao L, et al. A machine learning framework for identifying influenza pneumonia from bacterial pneumonia for medical decision making[J]. Journal of Computational Science. (iii) Li X, Fu Q, Li Q, et al. Multi-objective binary grey wolf optimization for feature selection based on guided mutation strategy. Applied Soft Computing. (iv) Qian L, Huang H, Xia X, et al. Automatic segmentation method using FCN with multi-scale dilated convolution for medical ultrasound image. The Visual Computer. Please ensure to cite.
Response 8: Studies from 2024 and 2025 have been added to the manuscript, and the suggested paper by Li et al. (Multi-objective binary grey wolf optimization for feature selection based on guided mutation strategy, Applied Soft Computing) has been cited accordingly. However, the other referenced papers, while valuable, are not closely related to the specific focus of our study and were therefore not included.
Comments 9: . Please elaborate on clinical implications of the current MAE/bMAE results. For instance, what would be an acceptable error margin from a clinical perspective?
Response 9: Clinicians typically evaluate Parkinson’s symptoms using discrete severity scores ranging from 0 to 4 (both in this study and in the MDS-UPDRS). An MAE or bMAE value below 1, therefore, indicates that the model’s predictions, on average, deviate by less than one severity level. This is clinically meaningful and suggests a level of precision that approaches—or even exceeds—that of standard clinical assessments. However, it is important to note that these metrics represent averages across all samples; some predictions may have small errors, while others may exceed one severity level. When examining class-specific MAE values (bMAE), we observe that for tremor and stiffness, an error greater than 1 occurs only at the highest severity level (4). For bradykinesia, it is exceeded at severity levels 3 and 4. This suggests that the models perform adequately for lower severity levels but require further refinement—and likely more training data—to improve accuracy in detecting the most severe cases. Nevertheless, this level of accuracy is promising, as it implies that the model could reliably support clinical decision-making by providing symptom severity estimates that are at least as consistent as those from clinician ratings.
Round 3
Reviewer 2 Report
Comments and Suggestions for Authors
From the response letter, the paper has been well revised, and the current version of the manuscript is acceptable for publication.